# A Balanced Time Perspective and Burnout Syndrome in the Corporate World

**DOI:** 10.3390/ijerph192114466

**Published:** 2022-11-04

**Authors:** Olga Klamut, Lening A. Olivera-Figueroa, Simon Weissenberger

**Affiliations:** 1First Faculty of Medicine, Department of Psychiatry, Charles University, 12108 Prague, Czech Republic; 2Department of Psychiatry, School of Medicine, Yale University, New Haven, CT 06510, USA; 3Department of Psychology, University of New York in Prague, 12000 Prague, Czech Republic

**Keywords:** burnout syndrome, time perspective, balanced time perspective, occupational health, biopsychosocial model

## Abstract

Burnout syndrome is officially classified in the International Classification of Diseases as an occupational phenomenon resulting from chronic workplace stress. Each year it is having an increasingly negative impact on the mental and physical health of employees, as well as on health costs and business performance. With this study, we aim at verifying whether there is a greater propensity for burnout depending on an individual’s time perspective, based on the framework of Christina Maslach’s burnout syndrome theory (consisting of three burnout dimensions), and Phillip Zimbardo’s Time Perspective (consisting of five distinct temporal profiles). Within the time perspective construct, we focused on an indicator of temporal adaptation, referred to as a Balanced Time Perspective (BTP). We used the Maslach Burnout Inventory and the Zimbardo Time Perspective Inventory on a sample of 129 Polish corporate employees. We found that two dimensions of burnout (emotional exhaustion and feelings of personal achievement) were significantly correlated to a balanced time perspective, while the third (depersonalization) did not pose a significant correlation. This underlines the interrelationships between personality and burnout, which gives way to one possible solution towards the danger of burnout syndrome—balancing an individuals’ time perspective through measures such as Time Perspective Therapy. We believe that the awareness of one’s temporal profile gives way to supplement gaps in one time perspective, while deterring the excessive effects of another, resulting in a more balanced time perspective, greater mental health and protection from burnout syndrome.

## 1. Introduction

Burnout syndrome is increasingly a problem in the work environment and concerns a growing number of people, leading to a wide range of unhealthy and even life threatening physical and mental consequences. Research shows that workplace stress is associated with over 120,000 deaths per year and approximately 5–8% of annual healthcare costs in the United States alone. This amounts to between 125 to 190 billion USD every year in healthcare costs [1].

The reasons for burnout can be found both in external and internal factors. The external being the environment of an organization, the laws and regulations surrounding work, career development and attempts to deal with the requirements of the labour market and its prevailing competitiveness. Our research focuses on the internal factors, the possible personality dictating a susceptibility towards burnout, as well as the other side of the spectrum—the personality measures which protect and buffer against the negative consequences of workplace strain. We describe this within the framework of the theory of time perspective, a measure that has a verified impact on decisions, actions and ways of experiencing life. The concept of time perspective has rarely been applied to research within organisational psychology and workforce problems, such as burnout syndrome. It is conceptualized by Zimbardo and Boyd using the Zimbardo Time Perspective Inventory, from now on referred to as ZTPI. Time perspective theory implies that an individual develops a way of perceiving time based on five temporal profiles—past negative, past positive, present hedonistic, present fatalistic, and future. A temporal profile that is not balanced between the five perspectives, leads to an excess or deficiency in some areas, having its consequence in inadequate ways of coping and unhealthy actions in everyday life. Thus, we concentrate on the measure of a balanced time perspective, which is closely correlated to better mental health.

The attempt to correlate time perspective with burnout syndrome was inspired by the research-based conviction that awareness of one’s own time perspective profile leads to personal agency and flexibility. This results in the possibility to adapt to various external situations, which in consequence leads to better ways of coping with them. While it might take a while to see substantial changes in these external factors leading to burnout, we believe that there is always the possibility to adjust accordingly with one’s own resilience, awareness and self-regulatory mechanisms.

In this paper, we present research conducted on 129 corporate employees in Poland. The took the Zimbardo Time Perspective Inventory and the Maslach Burnout Inventory to assess time perspectives as well as level of burnout. Besides the basic time perspective (TP) and burnout measures, we additionally divided our participants between those working less than and over 5 years, to see if working in one position also had a statistical influence worth noting. Three distinct hypotheses were formed to specify the relationship between a balanced time perspective and the susceptibility to burnout, two of which were found to be statistically verified. We sum up our findings with a focus on practical implications and where to go next in terms of slowing down the perpetual cycle of burnout syndrome in the workplace and increasing the awareness of burnout syndrome’s impact on public health.

## 2. Burnout Syndrome

Burnout syndrome is a topic that has grown in interest by various medical and psychological associations in recent years, and in 2019 was added by the World Health Organisation to the International Classification of Diseases (ICD-11), a diagnostic manual used by mental health professionals worldwide. The official ICD definition is the following:

“A syndrome conceptualized as resulting from chronic workplace stress that has not been successfully managed. It is characterised by three dimensions [2]:Feelings of energy depletion or exhaustionIncreased mental distance from one’s job or feelings negative towards one’s careerReduced professional productivity”

The severity of burnout syndrome has been further impacted by the recent SARS-CoV-2 pandemic. According to a recent study conducted by Flexjobs [3], workers are now more than three times as likely to report poor mental health than they were before the pandemic. Additionally, 40% have experienced burnout during the pandemic alone.

Increasing demands placed on employees, modern socio-economical changes, an increase in the competitiveness of the labour market, as well as the stress associated with work and maintaining it, are all contributing factors to the increasing prevalence of burnout syndrome [4].

It is important to differentiate between long-term stress and burnout syndrome, which are distinctly different phenomena. The immediate effects of long-term stress include discouragement, psychological exhaustion, and a withdrawal from work-related activities [5]. In turn, Stanisława Tucholska [6] proposes the following categories of burnout symptoms: emotional, cognitive, somatic, behavioural and motivational. Affective symptoms include depressive states, exhaustion, anxiety, a decrease in emotional control, irritation, decreased empathy and an increase in aggression, as well as a general lack of work satisfaction. Symptoms falling into the cognitive category are feelings of helplessness and powerlessness, low self-esteem, memory and concentration deficits, rigidity of thinking, hostility, suspicion, a severity of the projection mechanism, and a lack of trust towards colleagues and authorities in the workplace. Behavioural symptoms include acting on impulse, the inability to rest, isolation, a decrease in the quality of one’s work, being reluctant to new ideas, absences, as well as avoiding time spent at work and in contact with colleagues. Motivational aspects of burnout syndrome are primarily a loss of values such as a sense of meaning associated with work, high expectations and ambitions, work satisfaction, initiative and creativity, as well as a general willingness to work. Burnout syndrome also has its symptoms within somatic dispositions such as head and back aches, sleep disorders, sudden body weight changes, cardiac and gastric disorders, hypertension, and a general weakening of the immune system [6]. People who suffer from burnout lose sense of their work because of a prolonged reaction to emotional, physical and mental exhaustion, and they are unable to meet the requirements and demands of their jobs and work environments [7]. This exhaustion factor is the primary difference between long-term stress and burnout syndrome. Burnout syndrome is at the very end of the stress continuum. When chronic stress has not been managed and released through the body, it causes severe symptoms resulting in an impaired ability to function [8].

Having reviewed the detailed symptoms of burnout, we focus our research on the Maslach Burnout Inventory (MBI), which has been most widely used in research since its publishing in 1981. The MBI aligns with the World Health Organisations classification of burnout and that of the ICD-11 by regarding it within three components: emotional exhaustion (EE) focusing on feelings of being emotionally overextended, depersonalization (DP) meaning having an impersonal response towards recipients of one’s services, and personal accomplishment (PA), which translates to feelings of competence in one’s work [9].

## 3. Time Perspective

### 3.1. Zimbardo Time Perspective Inventory (ZTPI)

Time perspective is an individual’s attitude towards time, which is influenced by personal experiences and subject to cultural differences [10]. Especially in the context of occupational health, time is seen as an irreplaceable human resource. The time perspective theory coined by Zimbardo and Boyd, as well as over a decade of research in the field, led to the creation of a promising, multidimensional approach that operationalizes an individual’s relationship to time. The authors created a questionnaire, the Zimbardo Time Perspective Inventory (ZTPI), which allows for the characterization of people based on their relationship with time. They list five ways of recognizing of time-temporal orientations. Each shows a different hierarchy of values and different emotional attitudes to each other and the outside world, as well as various ways to respond with specific behaviours to a variety of life’s situations. The division of time perspectives are past positive, past negative, present hedonistic, present fatalistic and future.

Researchers presented convincing evidence that the use of such an individualized approach to the past, present and future, leads to measurable developmental results in fields such as academic performance [11], risky behaviours [12], stimulant use [13], as well as the frequency of physical activity [14]. Future time perspective is mainly associated with a high level of internal control and a general positive affect [15]. A past positive time perspective highly correlates with greater self-esteem, a perceived sense of safety, as well as higher levels of amicability [16]. Moreover, research has shown that people who are more past positive, tend to be more emotionally intelligent [17]. Oppositely, past negative time perspective is highly correlated with a tendency towards depression, lower self-esteem, problems with forming social relations, as well as a tendency towards addictions of all sorts [18]. The present hedonistic time perspective is highly correlated with a tendency towards risky behaviours, such as speeding, excessive drinking or drug use [19]. It is quite reasonable that present fatalist and past negative time perspectives are not beneficial for one’s health. Research shows that a high bias of these time perspectives is significantly related to low levels of self-realization and lower positive expectations [20].

The above examples are just a few portraying that a bias of a specific time perspective profile leads to specific lifestyle behaviours. This tendency usually leads to a significant narrowing of the temporal perspective to only one of the component’s traits, which in consequence becomes a relatively constant predisposition. This, in turn, can predict how an individual will act and cope with various situations. This type of narrowing often has to do with a series of negative consequences, which are usually associated with negative stress coping strategies and unfavourable life conditions. In search of an answer to the question of which time perspective is healthiest, Zimbardo and Boyd proposed a separate temporal scheme—the balanced time perspective.

### 3.2. Balanced Time Perspective—BTP

A balanced time perspective is a combination of high results in past positive, moderate results in future and present hedonist, alongside low results in past negative and present fatalist. Such an integration of time perspective specific personality traits and behaviours is theoretically said to correspond to mental and physical health, as well as proper social functioning. Behaviours and traits that are representative of a balanced time perspective are, amongst others, a sense of ambition towards future goals, optimism, reasonable diligence, a natural tendency to see the consequences of one’s behaviour, emotional intelligence, a high level of inner control, as well as believing in one’s own self-sufficiency, and a reasonable self-esteem. Further characteristics include a low sense of fear, a low inclination towards depressive states and a lack of problems with stress coping [21].

The past couple of years have shown a significant growing interest with the subject of balanced time perspective, which led to multiple empirical studies being conducted regarding BTP and various personality components and psychological phenomena. Significant correlations between balanced time perspective were found in an interaction with life satisfaction [22], a sense of happiness and mindfulness [23], higher levels of emotional intelligence [17], as well as cortisol levels [24]. An excess in any of the five temporal perspectives is similar to getting stuck in one dimension of action and decisions, in other words—a very narrow way of perceiving reality through only one scope of reasoning. The most beneficial for our mental health and the versatility of our actions is to consciously work towards achieving and maintaining a balanced time perspective. BTP is a measurable tool that allows to see where one is on the spectrum and to track progress in becoming more balanced. A popular and standardised method of measuring BTP is that of the deviation from a balanced time perspective—DBTP. It shows the deviation from an optimal time perspective profile. In a recent systematic review assessing the current empirical literature regarding DBTP, Stolarski et al. assessed 49 studies focusing on relationships between DBTP and psychological phenomena such as mental health, cognition, personality traits and biological correlates [25].

## 4. Materials and Methods

### 4.1. Research Aims and Hypotheses

The comprehensive goal of the study was to attempt to determine whether there is a correlation between temporal orientation and a tendency towards burnout syndrome symptoms. Various previous research has shown strong correlations between temporal orientation and stress [24,26,27].

For the purpose of this research, the assessment of whether a person identifies with the characteristics of burnout performed through the prism of the three burnout syndrome components, according to the theory of Christina Maslach: emotional exhaustion, depersonalization and feelings of personal achievements. Three research hypotheses were formed based on these components:
**Hypothesis** **1** **(H1).***Emotional exhaustion will be higher with a lower level of DNTP.*
**Hypothesis** **2** **(H2).***Depersonalization will be higher with a lower level of DNTP.*
**Hypothesis** **3** **(H3).***Feelings of personal achievement will be higher with a higher DNTP.*
where DNTP—deviation from negative time perspective, DBTP—deviation from balanced time perspective. Both will be thoroughly explained in further parts of the paper.

### 4.2. Sample and Research Procedure

The sample size was 129 people (N = 129), consisting of 74 female and 55 males. The age groups were the following: 48 people aged 20–29, 56 people aged 30–49 and 25 people aged 50+. The participants had to meet the requirement of being employed. The sample was divided into subgroups depending on their seniority-working over 5 years (N = 88), and less than 5 years (N = 41). Participants were of various corporate professions, working in large corporations based in Wrocław, Poland (such as Credit Suisse, Kaijima, Volvo, Google).

At the initial stage of the study, which consists of two psychometric questionnaires, a few socio-demographic questions were posed in order to determine the gender, age, job title and time working in specific position.

### 4.3. ZTPI—Zimbardo Time Perspective Inventory

To measure the participants individual profile of experiencing time, we used the 15-question shortened version of the ZTPI questionnaire, adapted to the Polish language [28]. The selection of the abbreviated version is justified by the participant’s better approach towards the study, as people usually prefer a shorter test time. The shorter version was verified in various correlating studies or time perspective with other variables, such as life satisfaction [29].

The Zimbardo Time Perspective Inventory locates an individual on a range of five subscales, each of which corresponds to the time perspective theory that is divided into five temporal dimensions. A factor analysis allowed the authors of the ZTPI to seclude the following subscales: past negative, past positive, present hedonistic, present fatalist and future [10]. Their internal consistency was measured by the Cronbach Alpha (comprising in the range from 0.00 to 1.00), which indicates the following for each subscale of the shortened version: present hedonism—α = 0.45; past positive—α = 0.54; present fatalist—α = 0.60; past negative—α = 0.78; future—α = 0.80. The value of the coefficient α of at least 0.70 is a satisfactory score (Cronbach, 1951) [30].

Table 1 shows examples of some of the questions in the questionnaire:

Besides showing where an individual is placed on each of the subscales, two single-variable components of time perspective were taken into account—balanced time perspective and negative time perspective. These can be obtained by comparing the obtained result with the optimal level on each of the subscales. Yet, for the purpose of this research, the results were described as one-dimensional, by calculating the deviation from the balanced time perspective [29]. The coefficient resulting from this deviation is known as the balanced time perspective (BTP) indicator.

The coefficient is calculated using the following formula:DBTP = √[(oPN-ePN)^2^ + (oPP-ePP)^2^ + (oPH-ePH)^2^ + (oPF-ePF)^2^ + (oF-eF)^2^]
whereas o = optimal level (PP = 4.60; PN = 1.95; PF = 1.50; PH = 3.90; F = 4.0) [31], e = empirical level (individual obtained result), and the rest of the abbreviations reflect the subscales (PN past negative, PP past positive, PH present hedonistic, PF present fatalist, F future).

The DBTP measures the difference between an individual’s time perspective, and the optimal time perspective profile (otherwise known as the balanced time perspective). The basis for appointing a balanced time perspective is the conviction that there is an optimal score on each of the time perspective subscales. The main BTP determinant is the closeness of the empirical result to the optimal one. In other words, the greater the DBTP, the greater the deviation from the balanced time perspective. Moreover, the DBTP being close to zero indicates a balanced time perspective and, therefore, greater mental health and wellbeing.

To further conceptualize results, a deviation from a negative time perspective [21] was also calculated. The greater the value of the DNTP is from zero, the more balanced the time perspective profile. The DNTP is defined as follows:DNTP = √[(nPN-ePN)^2^ + (nPP-ePP)^2^ + (nPF-ePF)^2^ + (nPH-ePH)^2^ + (nF-eF)^2^]
where, similarly to the DBTP formula; n = observed negative value for each TP, e = expected negative value (PN = 4.35; PP = 2.80; PF = 3.30; PH = 2.64; F = 2.75) [32].

Instead of placing an individual on five separate scales, each of them corresponding to different personality characteristics, only one value is analysed, the deviation from the balanced or negative time perspective profile. This type of reasoning allows for a one-dimensional comparison with other variables, simplifying the process and results. It is important to note that we used these two measures interchangeably, as a smaller measure of DBTP is an indication of better balance, while the opposite is true for DNTP (a greater measure indicates greater balance) [33]. Our statistical analysis only showed a statistical significance between the variables and DNTP, which is discussed in the limitations.

### 4.4. MBI—Maslach Burnout Inventory

Individual susceptibility of an individual towards the risk of burnout was examined by the Maslach Burnout Inventory. The third version of the questionnaire was used, which is directed towards all professions. The tool scores high in accuracy and reliability [34] due to its frequent use in research over the past 41 years, as well as various studies carried out that support its internal reliability. Findings report Cronbach Alpha ratings of 0.90 for emotional exhaustion, 0.76 Depersonalization, and 0.76 for Personal accomplishment [35].

The 22 item MBI concentrates on the individual’s personal approach to their professional work, as well as their feelings associated with work. The full inventory can be found in Appendix A.

Each item is assigned to one of the three —exhaustion, depersonalization and feelings of personal achievement. Similarly to the ZTPI, each subscale corresponds to the theoretical components of burnout. Responses to affirmative sentences are placed on a 7-point scale regarding prevalence of the feelings (0—never, 1—several times a year or less frequently, 5—several times a week, and 6—every day).

The result is determined by summing up the results obtained for the individual subscales, as shown in Table 2:

For the subscales of “emotional exhaustion” and “depersonalization”, the level of burnout is greater with the higher score. In turn, for the subscale “personal achievements”, the lower the score, the greater the degree of burnout.

## 5. Results

All accumulated results were analysed using the SPSS statistical program. Besides the socio-demographic, the following variables were taken into statistical analysis:Three components of burnout: emotional exhaustion, depersonalization, feelings of personal achievementFive time perspectives profiles: past negative, past positive, present hedonist, present fatalist, futureDeviation from a balanced time perspective DBTP, deviation from a negative time perspective DNTP

The basic correlations between time perspective and burnout components are presented in Table 3. The results marked with a star (*) account for a statistical significance on the level of 0.01; and results marked with two stars (**) show a significance on the 0.05 level. The results were calculated by implementing Pearson’s Correlation, which identifies the level of linear dependence between variables. Additionally, Table 4 shows the foundation statistics for each variable—being the minimum, maximum, average and standard deviation.

Additionally, the sample was divided into two subgroups—individuals working less than 5 years and over 5 years in their current position. The basic statistical calculations did not show significant differences in the results of the burnout components, shown in Table 5. Yet it is worth noting that overall scores in the depersonalization category were slightly higher for those working less than 5 years in one job position; therefore, these were usually younger people. Exploring reasons to this may be an interesting starting point for further research; yet, in terms of this work, it has no noteworthy influence on time perspective.

### 5.1. Linear Regression

The primary statistical analysis shows correlations between variables whose relationship needs to be further examined. For this reason, interrelationships are shows in graphs with linear regression parameters, which indicate the direction of correlation between the dependent and independent variables. This type of analysis was specified for each of the three burnout components, DBTP and DNTP, which are depicted in Table 6, Table 7 and Table 8.

The results proved statistically significant correlations with DNTP (deviation from negative time perspective) for two of the three burnout components—emotional exhaustion and personal achievements. Table 9 shows their directions:

### 5.2. Emotional Exhaustion

Results show that DNTP has a negative correlation with emotional exhaustion (on the level of −0.444). This signifies that the greater the deviation from a negative time perspective profile, the lower the emotional exhaustion results. This indicates the first hypothesis to be true (H1: Emotional Exhaustion will be greater with a lower result of DNTP). Figure 1 shows the distribution of results using a linear regression analysis.

### 5.3. Personal Achievement

In the case of the personal achievement variable, the third hypothesis was also supported. DNTP predicts the growth of personal achievement feelings (H3: Feelings of personal achievement will be higher, with a higher level of DNTP), as shown in Figure 2. This signifies that the further an individual is from a negative time perspective profile, the greater their feelings of personal achievements will be.

### 5.4. Depersonalization

The linear regression model in case of the depersonalization variable is *p* = 0.097, which indicates the model to be marginally significant, yet statistically insignificant. This marks the second hypothesis as invalid. (H2: Depersonalization is expected to be greater with a lower result of DNTP). Yet the correlation itself between DNTP and depersonalization results in *p* = 0.037, which indicates that DNTP predicts decreased depersonalization in a marginally significant regression model. The conclusion in this case can be made that DNTP does indeed have an influence on depersonalization in the case of burnout syndrome, yet an official statement cannot be made in terms of statistical significance.

## 6. Discussion

The statistical results of our study are the following:oAn increased DNTP predicts a lower level of emotional exhaustion (*p* = 0.000; beta = −0.444; t = −5.045; C.I = −8.445–(−3.686));oAn increased DNTP predicts higher levels of personal achievement (*p* = 0.000; beta = 0.502; t = 5.780; C.I = 3.114–6.357);oIn contrast, DNTP is not statistically correlated with depersonalization (*p* = 0.097; beta = −1.553; t = −2.104; C.I = −3.014–(−0.092)).

The results apply to our research hypothesis in the following manner:

**Hypothesis** **1** **verified** **(H1** **verified).**
*Emotional exhaustion will be higher with a lower level of DNTP.*


**Hypothesis** **2** **not** **verified** **(H2** **not** **verified).**
*Depersonalization will be higher with a lower level of DNTP.*


**Hypothesis** **3** **verified** **(H3** **verified).**
*Feelings of personal achievement will be higher with a higher DNTP.*


Therefore, we can conclude that two out of the three burnout syndrome components are correlated to a deviation from a negative time perspective and, therefore, to an overall balanced time perspective. The attained results confirm what could be deemed as intuitive, and what the Zimbardo time paradox concerns—that a balance of positive characteristics from each time perspective leads to a healthy psychosomatic lifestyle, which in consequence is a determinant of mental wellbeing. The awareness of one’s temporal profile can lead to supplement gaps in one time perspective, while deterring the excessive effects of another. Apart from embodying a balanced time perspective, an individual can work on developing the ability to flexibly move between time perspectives, adjusting their behaviour to certain situations and, in turn, assimilating and coping in healthier ways.

## 7. Conclusions

Our study showed that burnout syndrome is closely tied to personality measures and that an individual’s temporal profile plays an important role in their susceptibility towards having the symptoms that burnout includes. Our findings are based on the concept of a balanced time perspective, which is a precursor of greater mental health and results from a balance of the five temporal profiles according to Zimbardo’s time perspective theory: past positive, past negative, present hedonistic, present fatalistic and future time perspective. The results of our research showed that two out of the three burnout syndrome dimensions are tied to a balanced time perspective. The scale of emotional exhaustion and feelings of personal achievement are statistically correlated with a balanced time perspective profile, while the dimension of depersonalisation was not found to be statistically relevant, which is an important scope of research to follow up on in future research.

In the Flexjobs survey on employee mental health and burnout syndrome that we previously mentioned, 56% of respondents stated that having flexibility in their workday would be the best way in which their employers could support them. While implementing external changes in the biopsychosocial work environment is an important step to take towards greater health and functioning of employees and their companies, what if employees could grant themselves this flexibility on an internal level? One that is present as an embodied personality measure regardless of external circumstances, which, as the past few years in the workplace have shown, are very subject to change. We encourage to turn the focus of occupational health onto how building inner flexibility and a balanced time perspective is a beneficial trait that oversees external circumstances, such as the wellbeing structure of the workplace. Using temporal orientation as a tool towards self-regulation and implementing healthy coping strategies allows for individuals to have agency over their decisions, reactions, and beneficially adjust to workplace requirements.

A possible way of working with these results could be by means of Time Perspective Therapy (TPT), an evolution of Cognitive Behavioural Therapy (CBT). It has so far been mostly implemented on individuals suffering from PTSD. Research shows that after 4–8 meetings, on average, individuals experiencing PTSD symptoms had less anxiety and depressive states, on a statistical significance level of 0.001 [36]. The aim of TPT is primarily to identify the five time perspectives of a patient, to assess their levels and to create a time perspective profile based on the deviations from balanced and negative profiles. With this knowledge, the following step would be tailoring one’s profile towards a more balanced one. This is achieved by mindset work and learning self-regulatory tools to further balance each profile. Such a method could be implemented in the workplace, aimed at improving employees stress coping strategies and effectively dealing with outside factors that could potentially put them in danger to burnout in the long run. The results of research on this specific type of therapeutic support confirm the fact that being aware of our own time perspective could be very beneficial to dealing with problems at work, as well as to being more flexible and more proactively adjusting to the requirements of various circumstances in life. Companies are continuously becoming more open towards wellness practices and mental health awareness [36,37,38,39] and the global corporate wellness market is expected to be valued at $77 billion by 2027 [40]. Based on our findings, we suggest that the topic of time perspective be included in this conversation, to expand the improvement of employee’s mental health beyond workplace environment and into building personality measures that will protect employees from the dangers of burnout syndrome at work and beyond.

## 8. Limitations

Our research undoubtedly has limitations that must be considered. First, all the information was self-reported by the study’s participants, who were also not varied geographically or culturally. While our study shows that time perspective can be considered as a predictor towards the susceptibility of burnout syndrome, more work is needed around the validity of the DBTP and DNTP measurements. In a recent study also comparing TP and burnout tendencies, Unger et al., [41] chose to focus only on DNTP, since its low score has a higher burnout risk due to implying high scores in present fatalistic and past negative. We agree with this reasoning and more so support it with our results being statistically significant only in DNTP, and not in the DBTP measure. On the other hand, a different recent study focusing on burnout within blue collar workers [42] showed that DBTP has a significant influence on burnout proneness. The varying results around DBTP show that this measure needs to be undertaken in more research on varying groups in order to assess it concisely.

## Figures and Tables

**Figure 1 ijerph-19-14466-f001:**
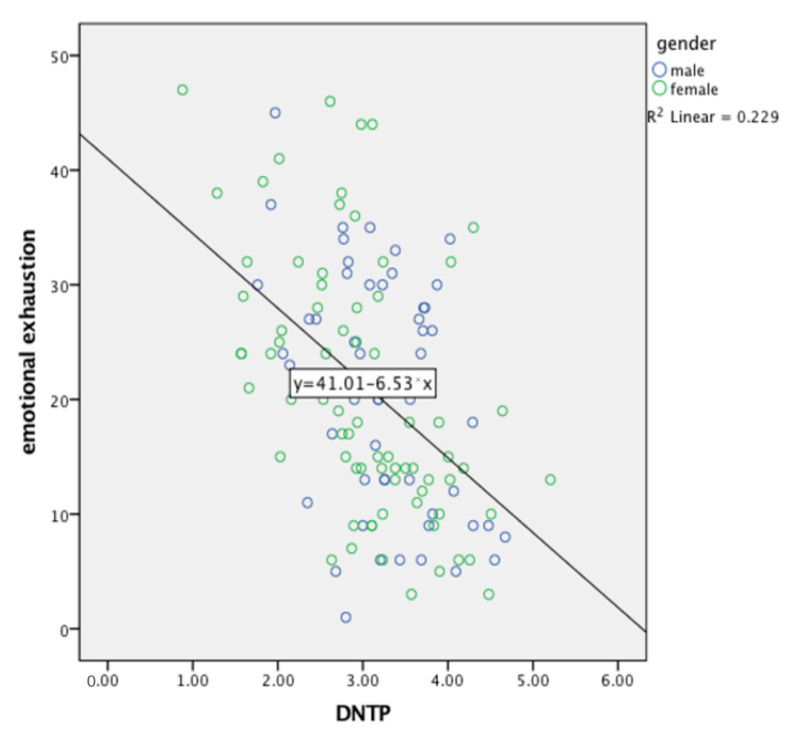
Linear regression analysis of DNTP and emotional exhaustion.

**Figure 2 ijerph-19-14466-f002:**
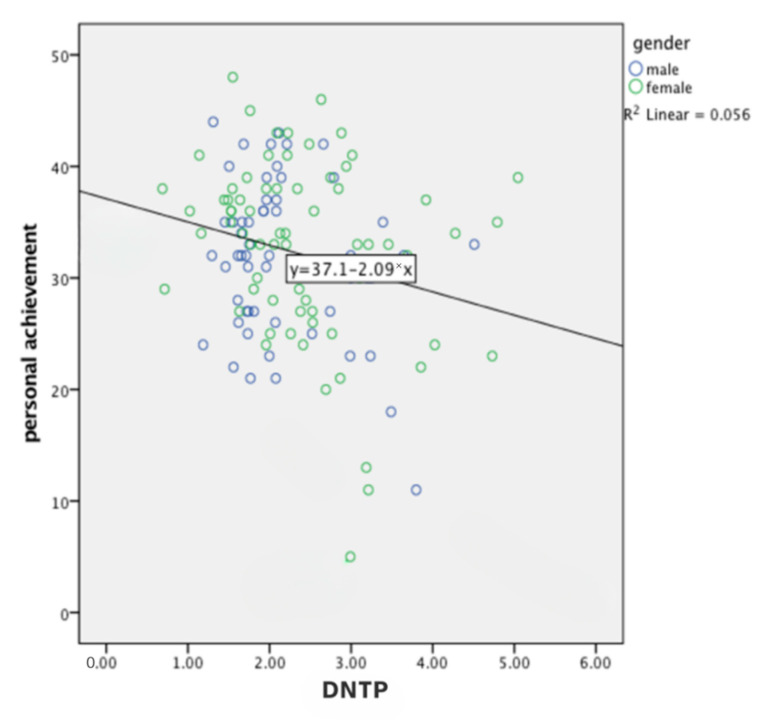
Linear regression analysis of DNTP and personal achievement.

**Table 1 ijerph-19-14466-t001:** Examples of questions from the ZTPI. Full questionnaire can be found in Appendix A.

Past Positive	Happy memories of good times spring readily to mind.
Past Negative	It is hard for me to forget unpleasant things from my past.
Present hedonistic	I believe that getting together with one’s friends to party is one of life’s important pleasures.
Present fatalist	What is supposed to be, will be, therefore the actions I take do not make much of a difference.
Future	I am able to resist temptations when I know that there is work to be done.

**Table 2 ijerph-19-14466-t002:** Ranges of results of the MBI.

Emotional Exhaustion	High result > 27	Average result 17–26	Low result 0–16
Depersonalization	High result > 13	Average result 7–12	Low result 0–6
Feelings of personal achievement	High result 0–31	Average result 32–38	Low result > 39

**Table 3 ijerph-19-14466-t003:** Basic linear correlations between variables.

	EE	D	PA	PN	PP	PF	PH	F	DBTP	DNTP
EE	1									
D	0.486 **	1								
PA	−0.355 **	−0.169	1							
PN	0.479 **	0.154	−0.342 **	1						
PP	−0.095	0.041	0.141	−0.179 *	1					
PF	0.189 *	0.155	−0.181 *	0.253 **	0.112	1				
PH	−0.113	0.017	−0.368 **	−0.120	0.308 **	0.056	1			
F	−0.059	−0.007	−0.180 *	−0.190 *	0.095	−0.023	−0.95	1		
DBTP	0.279 **	0.051	−0.237 **	0.659 **	−0.570 **	0.440 **	−0.157	−0.222 *	1	
DNTP	−0.478 **	−0.186 *	−0.504 **	−0.707 **	−0.294 **	−0.377 **	0.437 **	0.407 **	−0.464 **	1

EE: Emotional Exhaustion, D: Depersonalization, PA: Personal Achievements, PN: Past Negative, PP: Past Positive, PH: Present Hedonistic, PF: Present Fatalistic, F: Future, DBTP: Deviation from Balanced Time Perspective, DNTP: Deviation from Negative Time Perspective.

**Table 4 ijerph-19-14466-t004:** Basic statistic calculations for N = 129.

N = 129	Minimum	Maximum	Average	Standard Deviation
Emotional Exhaustion	1	47	20.7	10.875
Depersonalization	0	26	6.73	5.959
Personal Achievements	5	48	32.22	7.517
Past Negative	1.00	5.00	2.7780	0.99703
Past Positive	1.33	5.00	3.3486	0.75208
Present Fatalist	1.00	5.00	2.2609	0.85085
Present Hedonist	1.67	5.00	3.9871	0.68760
Future	1.33	5.00	3.9018	0.78572
DBTP	0.69	5.04	2.3391	0.85414
DNTP	0.88	5.21	3.1119	0.79672

**Table 5 ijerph-19-14466-t005:** Basic statistical calculations for individuals working less than 5 years, N = 41, and more than 5 years, N = 88.

	Emotional Exhaustion	Depersonalization	Personal Achievements
<5 average	21.44	8.88	32.27
<5 standard deviation	11.598	7.201	5.805
>5 average	20.35	5.73	32.19
>5 standard deviation	10.572	5.021	8.224

**Table 6 ijerph-19-14466-t006:** Linear regression of Emotional Exhaustion.

(A) DNTP, DBTP and Emotional Exhaustion												
Model Summary ^b^												
					Change Statistics							
R	R Square	Adjusted R Square	Std. Error of the Estimate		R. Square Change	F Change	df1	df2	Sig. F Change		Durbin- Watson	
0.482 ^a^	0.233	0.221	9.601		0.233	19.108	2	126	0.000		2.009	
^a^ Predictors: (Constant), DNTP, DBTP ^b^ Dependant Variable: emotional exhaustion												
Coefficients ^a^												
	Unstandardized Coefficients		Standardized Coefficients Beta	t	Sig.	95.0% Confidence Interval for B		Correlations			Collinearity Statistics	
	B	Std. Error				Lower Bound	Upper Bound	Zero Order	Partial	Part	Tolerance	VIF
(Constant)	37.409	5.541		6.752	0.000	26.444	48.373					
DBTP	0.925	1.121	0.73	0.825	0.411	−1.294	3.145	0.279	0.073	0.064	0.785	1.274
DNTP	−6.066	1.202	−0.444	−5.045	0.000	−8.445	−3.686	−0.478	−0.410	−0.394	0.785	
^a^ Dependant Variable: emotional exhaustion												

**Table 7 ijerph-19-14466-t007:** Linear Regression of Depersonalization.

(B) DNTP, DBTP and Depersonalizatio												
Model Summary ^b^												
					Change Statistics							
R	R Square	Adjusted R Square	Std. Error of the Estimate		R. Square Change	F Change	df1	df2	Sig. F Change		Durbin- Watson	
0.191 ^a^	0.036	0.021	5.896		0.036	2.380	2	126	0.097		1.964	
^a^ Predictors: (Constant), DNTP, DBTP ^b^ Dependant Variable: depersonalization												
Coefficients ^a^												
	Unstandardized Coefficients		Standardized Coefficients Beta	t	Sig.	95.0% Confidence Interval for B		Correlations			Collinearity Statistics	
	B	Std. Error				Lower Bound	Upper Bound	Zero Order	Partial	Part	Tolerance	VIF
(Constant)	12.308	3.402		3.617	0.000	5.574	19.041					
DBTP	−0.319	0.689	−0.046	−0.463	0.644	−1.682	1.044	0.051	−0.041	−0.040	0.785	1.274
DNTP	−1.553	0.738	0.208	−2.104	0.037	−3.014	−0.092	−0.186	−0.184	−0.184	0.785	1.274
^a^ Dependant Variable: depersonalization												

**Table 8 ijerph-19-14466-t008:** Linear Regression of Personal Achievement.

(C) DNTP, DBTP and Personal Achievement												
Model Summary ^b^												
						Change Statistics						
R	R Square	Adjusted R Square	Std. Error of the Estimate		R. Square Change	F Change	df1	df2	Sig. F Change		Durbin- Watson	
0.504 ^a^	0.254	0.242	6.543		0.254	21.461	2	126	0.000		1.717	
^a^ Predictors: (Constant), DNTP, DBTP ^b^ Dependant Variable: personal achievement												
Coefficients ^a^												
	Unstandardized Coefficients		Standardized Coefficients Beta	t	Sig.	95.0% Confidence Interval for B		Correlations			Collinearity Statistics	
	B	Std. Error				Lower Bound	Upper Bound	Zero Order	Partial	Part	Tolerance	VIF
(Constant)	17.574	3.776		4.654	0.000	10.101	25.046					
DBTP	−0.040	0.764	−0.005	−0.052	0.958	−1.553	1.472	−0.237	−0.005	−0.004	0.785	1.274
DNTP	4.736	0.819	0.502	5.780	0.000	3.114	6.357	0.504	0.458	0.445	0.785	1.274
^a^ Dependant Variable: personal achievement												

**Table 9 ijerph-19-14466-t009:** Correlation direction from linear regression of statistically significant correlations.

	Correlation Direction
Emotional Exhaustion	−0.444
Personal Achievements	0.502

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
