# Peer review of "A Balanced Time Perspective and Burnout Syndrome in the Corporate World"

_ijerph, 2022, doi:10.3390/ijerph192114466_

Round 1

Reviewer 1 Report

1. Please provide the whole questionnaire as supplementary data.

2. I think some researches have a lot of content related to this study here, but the authors did not cite and discuss them. Eg. Psychological Studies volume 67, pages150–163 (2022), Int J Environ Res Public Health. 2020 Sep; 17(18): 6905.

3. Line242: Note that there are errors in the math notation.

4. Line229&244: (Zimbardo, Boyd, 2008) (Zimbardo, Sword, 2012). What are these messages? References? Please unify the format.

Author Response

Reviewer #1 comments and responses:

  1. Please provide the whole questionnaire as supplementary data.

Both the Zimbardo Time Perspective Inventory and Maslach Burnout Inventory have been added as supplements.

  1. I think some researches have a lot of content related to this study here, but the authors did not cite and discuss them. Eg. Psychological Studies volume 67, pages150–163 (2022), Int J Environ Res Public Health. 2020 Sep; 17(18): 6905.

 Both were added to the paper in the Limitations section:

‘Our research undoubtedly has limitations that must be considered. First, all the information was self-reported by the studies participants, who were furthermore not varied geographically or culturally. While our study shows that time perspective can be considered a predictor towards the susceptibility of burnout syndrome, more work needs to be done around the validity of the DBTP and DNTP measurements. In a recent study also comparing TP and burnout tendencies, Unger et al, [42] chose to focus only on DNTP, since it’s low score has a higher burnout risk due to implying high scores in present fatalistic and past negative. We agree with this reasoning and moreso support it with our results being statistically significant only in DNTP, and not in the DBTP measure. On the other hand, a different recent study focusing on burnout within blue collar workers [43] showed that DBTP has a significant influence on burnout proneness. The varying results around DBTP show that this measure needs to be undertaken in more research on varying groups in order to assess it concisely. ‘

  1. Line242: Note that there are errors in the math notation.

The equation has been corrected.

  1. Line229&244: (Zimbardo, Boyd, 2008) (Zimbardo, Sword, 2012). What are these messages? References? Please unify the format.

Yes, these are references that we missed and have been fixed into the unified format, thank you for noticing this.

Reviewer 2 Report

I appreciate the Maslach and Zimbardo work as your theoretical foundation

the words "Corporate World" in title should be capitalized

In 5.4 PersonalAchievement section: "proved to be correct"should be changed to "was supported". Theories can never be "proven" as that would connote that the findings have been replicated and would replicate 100% of the time

Author Response

Reviewer #2 comments and replies:

  1. I appreciate the Maslach and Zimbardo work as your theoretical foundation

Thank you! They are theories that compliment each other very well and we hope to base further research on them as well.

  1. The words "Corporate World" in title should be capitalized

Done.

  1. In 5.4 Personal Achievement section: "proved to be correct" should be changed to "was supported". Theories can never be "proven" as that would connote that the findings have been replicated and would replicate 100% of the time

All information around the hypotheses was changed from ‘proved’ to ‘supported’ and ‘indicated’.

Reviewer 3 Report

This study examined the relationship between burnout syndrome and an individual’s balanced time perspective, which was an interesting research question. While I was reading, I felt confused about several issues. Hopefully, the authors could address them better.

2. Burnout syndrome

On page 2, line 83, the authors defined burnout syndrome as “a syndrome…resulting from chronic workplace stress that has not been successfully managed.” On next page, line 96, the authors stated that “It is important to differentiate between long-term stress and burnout syndrome, which are distinctly different phenomena.” Then the authors listed more definitions regarding burnout, but did not illustrate clearly how to differentiate between long-term stress and burnout syndrome, or how these two constructs are related to each other.

4.1 Research aims and hypotheses

On page 5, DNTP was used in the hypotheses, but before that DBTP was used. What is the relationship between DNTP and DBTP? Were the authors using them interchangeably?

4.3&4.4

Psychometrics of the measurements should be reported, such as reliabilities.

5. Results

Table 3 & 4 could be combined into one table.

Please provide the whole linear regression results, including the F-test, % of variance explained and each independent variable’s test results.

The figure on page 12 was about DBTP, not DNTP as mentioned in the text.

As I mentioned above, I’m confused about the difference between DNTP and DBTP. Why didn’t the authors use DBTP in the analysis? Or was the results of DBTP not significant, but the results of DNTP were significant?

6. Conclusion

The authors should add a section of limitations.

Author Response

Reviewer #3 comments and responses:

Comment: This study examined the relationship between burnout syndrome and an individual’s balanced time perspective, which was an interesting research question. While I was reading, I felt confused about several issues. Hopefully, the authors could address them better.

We hope that some of the uncertainty was cleared up with our edits- there was definitely a need for a limitations sections as well as a clearer view of DNTP and DBTP. Thank you for addressing this. 

  1. On page 2, line 83, the authors defined burnout syndrome as “a syndrome…resulting from chronic workplace stress that has not been successfully managed.” On next page, line 96, the authors stated that “It is important to differentiate between long-term stress and burnout syndrome, which are distinctly different phenomena.” Then the authors listed more definitions regarding burnout, but did not illustrate clearly how to differentiate between long-term stress and burnout syndrome, or how these two constructs are related to each other.

We added a clarifying sentence at the end of this paragraph with another citation from a 2016 review article focusing primarily on the difference between chronic stress and burnout syndrome:

‘People who suffer from burnout lose sense of their work because of a prolonged reaction to emo-tional, physical and mental exhaustion, and they are unable to meet the requirements and de-mands of their jobs and work environments [7]. This exhaustion factor is the primary difference between long-term stress and burnout syndrome. Burnout syndrome is at the very end of the stress continuum. When the chronic stress has not been managed and released through the body, which causes severe symptoms resulting in an impaired ability to function. [8]’

  1. On page 5, DNTP was used in the hypotheses, but before that DBTP was used. What is the relationship between DNTP and DBTP? Were the authors using them interchangeably?

We used both measures interchangeably which we additionally explained here, adding information on the statistical insignificance of DBTP and further reference to the limitations:

‘Instead of placing an individual on five separate scales, each of them corresponding to different personality characteristics, only one value is analysed, the deviation from the balanced or negative time perspective profile. This type of reasoning allows for a one-dimensional comparison with other variables, simplifying the process and results. It is important to note that we used these two measures interchangeably, as a smaller measure of DBTP is an indication of better balance, while the opposite it true for DNTP (a greater measure indicates greater balance) [34]. Our statistical analysis only showed a statistical significance between the variables and DNTP, which is discussed in the limitations.’

  1. Psychometrics of the measurements should be reported, such as reliabilities.

Psychometrics added:

MBI: The third version of the questionnaire was used, which is directed towards all professions. The tool scores high in accuracy and reliability [31] due to it’s frequent usage in research over the past 41 years, as well as various studies carried out which support it’s internal reliability. Findings report Cronbach alpha ratings of 0.90 for emotional exhaustion, 0.76 Depersonalization, and 0.76 for Personal accomplishment [32].

ZTPI: A factor analysis allowed the authors of the ZTPI to seclude the following subscales: past negative, past positive, present hedonistic, present fatalist, future [9]. Their internal consistency was measured by the Cronbach Alpha (comprising in the range from 0,00 to 1,00), which indicates the following for each subscale of the shortened version: present hedonism – α = 0,45; past positive – α = 0,54; present fatalist–     α = 0,60; past negative – α = 0,78; future – α = 0,80. The value of the coefficient α of at least 0,70 is a satisfactory score (Cronbach, 1951) [30].

  1. Table 3 & 4 could be combined into one table.

Table 3 & 4 combined into onto table.

   5. Please provide the whole linear regression results, including the F-test, %   of variance explained and each independent variable’s test results.

Tables added as requested.

6. The figure on page 12 was about DBTP, not DNTP as mentioned in the text.

Fixed.

7. As I mentioned above, I’m confused about the difference between DNTP and DBTP. Why didn’t the authors use DBTP in the analysis? Or was the results of DBTP not significant, but the results of DNTP were significant?

Only the results of DNTP were statistically significant, which we mentioned in section ‘5.2 Linear Regression’:

 The results proved statistically significant correlations with DNTP (deviation from negative time perspective) for two of the three burnout components- emotional exhaustion and personal achievements. Their directions are as follows:’

We further elaborated on this in the Limitations section, described below.

8. The authors should add a section of limitations.

‘Our research undoubtedly has limitations that must be considered. First, all the information was self-reported by the studies participants, who were furthermore not varied geographically or culturally. While our study shows that time perspective can be considered a predictor towards the susceptibility of burnout syndrome, more work needs to be done around the validity of the DBTP and DNTP measurements. In a recent study also comparing TP and burnout tendencies, Unger et al, [42] chose to focus only on DNTP, since it’s low score has a higher burnout risk due to implying high scores in present fatalistic and past negative. We agree with this reasoning and moreso support it with our results being statistically significant only in DNTP, and not in the DBTP measure. On the other hand, a different recent study focusing on burnout within blue collar workers [43] showed that DBTP has a significant influence on burnout proneness. The varying results around DBTP show that this measure needs to be undertaken in more research on varying groups in order to assess it concisely. ‘

Round 2

Reviewer 3 Report

I appreciate the efforts that the authors have made to address my comments.